# CONTINUOUS SURFACE NORMAL INTEGRATION

## ABSTRACT

We address a novel task for monocular explicit surface reconstruction that extends traditional surface normal integration over measurements on a regular grid to direct continuous surface depth estimation. Our solution accepts coordinates as queries and predicts both the normal and depth of an arbitrary query point by its relative locations and orientations to the points distributed in its vicinity. In general, all points are regarded by our model as random samples drawn from an underlying continuous gradient field of a surface which we parameterize using a field of polynomials to establish its topology. We establish a mapping from coordinates to a sequence of learnable polynomial coefficients to model a continuous surface and train a neural network to approximate it. We decompose a continuous surface representation into two components: (1) a set of grid points of unknown orientations whose locations are picked by a quadtree and (2) a set of sample points whose orientations are directly observable. Our training workflow estimates the normal of grid points and the locations of depth discontinuities iteratively. During each iteration, we generate a normal map of grid points for it to be processed by a standard bilateral normal integrator to identify the locations of depth discontinuities, which we use to refine the estimation for grid-based normal map in the subsequent iteration. As a result, the learned model generates both normal and depth for arbitrary coordinates accurately in a continuous field. We provide both theoretical formulation for our design and extensive empirical evidence to demonstrate that our proposed method not only delivers a performance as effective as its grid-based counterpart approaches but also flexibly and accurately addresses the continuous cases that existing methods are unable to handle.

## 1 INTRODUCTION

Normal integration establishes an inverse mapping from a surface's normal map to its depth. It completes the production cycle of multiple important 3d computer vision tasks including photometric stereo, shape from shading, etc., which settles on surface normal as their output. Most existing solutions to normal integration formulate the problem as an inverse problem of recovering a discrete scalar field from its corresponding gradient map by a numerical solver to the corresponding partial differential equation (PDE) in a 2D space subject to various boundary conditions. This PDE is often solved by a large linear system involving spatial numerical differentiation and recent research effort have been put into modeling and identifying the locations of depth discontinuities properly. In the discrete domain, PDE solvers consume input stored over a regular grid Quéau et al. (2018a), where the spacing between adjacent measurements is uniform so that numerical differentiation is properly defined consistently over the entire integration domain. For two reasons we believe it is necessary to develop a tool for explicit surface reconstruction that directly interoperates with continuous representations: on the device end, present vision data acquisition techniques may provide **unstructured** representations Peers et al. (2006) including surface normal which is directly made available for a wide spectrum of downstream tasks Xiu et al. (2023); on the design end, as **dense** representations for surface normal becomes an inalienable product of data-driven inverse rendering pipelines Bae & Davison (2024); He et al. (2024), but a model that explicitly and flexibly abridges dense representations involving surface normal and surface structure still remains elusive.

We study a new type of task for monocular explicit surface reconstruction and introduce a design that directly accepts coordinate-based queries and produces estimates of both depth and normal for query points of arbitrary coordinates. We regard the observed surface points as a set of random samples

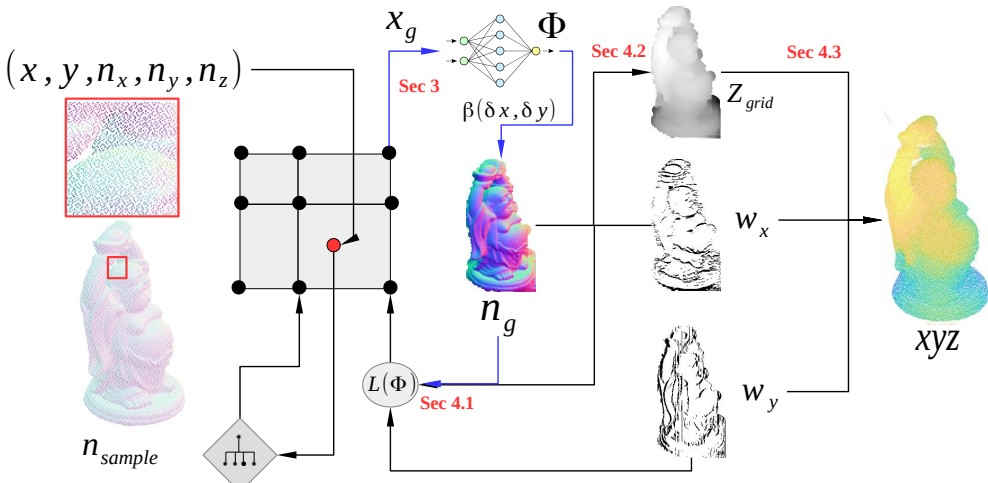

Figure 1: Our proposed design takes a set of unstructured input and establishes a grid of points, $x_g$ with locations specified by a quadtree. This grid of field is overlayed on a continuous field of polynomials, where through mapping approximated by a neural network, $\Phi$, we obtain the corresponding learnable polynomial coefficients, $\beta$, which allows us to setup the a normal map for $x_g$ (Section 3). In addition to depth map $Z_g$, This learned normal map is processed by a bilateral normal integrator also to deliver the location-wise estimate for non-differentiable depth discontinuities along $x$-axis and $y$-axis ($w_x$ and $w_y$), respectively. The updated information about differentiability can be used to refine the loss function training the network. Moreover, $w_x$ and $w_y$ can also be used to derive from $Z_g$ to obtain the depth estimate for arbitrary query points in the form of point cloud $xyz$. Data paths passing gradients are colored in blue.

drawn from an underlying continuous surface gradient field which we parameterize by polynomials Cazals & Pouget (2005). To this end, we establish a learnable mapping from the coordinates of query points to a sequence of polynomial coefficients to describe the geometry of a point-wise differentiable surface. We introduce a training pipeline to train a neural network that approximates this mapping. Essentially, because fitting surface by polynomials imposes pair-wise constraints between two connected points in terms of their respective local spatial gradients, this naturally leads to a loss function for this neural network to be trained.

In a more general setting, non-differentiable depth discontinuities exist that often prevent a single polynomial from fitting two points separated by the discontinuity consistently. Because Bilateral normal integration Cao et al. (2022) over a regular grid provides effective location-specific likelihood estimate for depth discontinuities in terms of the weight assigned to the edges between two connected grid points, to extend its benefit to the continuous domain, we decompose a continuous surface representation into two components: (1) a set of points on a non-regular grid whose orientations are to be estimated, and their connectivity is determined by a quadtree Samet (1984). (2) a set of sample points with observed surface orientations distributed around the grid points. Accordingly, there are two types of pair-wise constraints formed by two connected points in terms of how these two points located geometrically: (1) grid-grid connection, whose bonding is directly determined by a bilateral normal integrator in terms of a 0-1 weighting; (2) grid-sample connection: the connection between a grid point and its neighboring sample points whose weighting is derived from the nearby grid-grid connection.

Therefore, accurate depth map estimate for grid points is essential for continuous surface normal integration, and this depends on accurate estimation of the corresponding normal map, which is directly affected by how well the underlying field of polynomials fits it. Hence, including weighting of pair-wise point connections in a differentiability-aware loss function is critical for training a neural network that accurately approximates the mapping from field coordinates to the polynomial coefficients. This implies that the training has to carry out an iterative procedure that alternatively

refines the estimate for grid-based normal map and for the locations of depth discontinuities that encode shape topology. Moreover, since all operations are coordinate-driven, our method essentially delivers an explicit reconstruction for a continuous surface. Figure 1 provides an overview of our proposed design.

To sum up, our contributions are as follows:

1. A design of computational framework based on coordinate-driven queries that generalizes the classical normal integration of data stored on a regular grid to measurements of continuous representations.

2. A model of a continuous field of learnable polynomials that represent continuous surface and a depth discontinuity aware loss function that facilitates its training.

3. Analysis of results obtained from extensive experiments demonstrating that our proposed method not only performs as effective as the existing method on data stored on regular grid but also delivers equally good performance on continuous data representations that existing methods fail to handle.

This paper is organized as follows: Section 2 gives an overview of the existing literature; Section 3 explains the polynomial-based formulation for continuous surface modeling. Section 4 introduces how the field of polynomial coefficients are trained iteratively and how the inferred locations of depth discontinuities are used to refine the quality of both normal and depth estimation. Section 5 analyzes and visualizes estimation results obtained processing data of both regular grid and continuous representations. Section 6 discusses future work and concludes this paper.

## 2 RELATED WORK

Existing literature addresses normal integration as an inverse problem of a everywhere differentiable surface solved by a PDE solver, while more recent work also focuses on depth discontinuity detection and preservation. On the other end of spectrum lies a separate line of work that studies normal estimation for unstructured point cloud. Our work investigates the properties of both.

### 2.1 FORMULATION

Essentially, PDE solvers delivers a solution that is expected to minimize following energy-based functional Horn & Brooks (1986):

$$\min_z \int_\Omega E(\partial_u z(u,v) - p(\boldsymbol{n})) + E(\partial_v z(u,v) - q(\boldsymbol{n}))dudv, \tag{1}$$

where $p = -\frac{n_x}{n_z}$, $q = -\frac{n_y}{n_z}$, $\partial_u z$ and $\partial_v z$ are associated with a regular grid. In order for the solver to be numerically stable, orthogonal constraint is introduced in the presence of large noise Zhu & Smith (2020). Moreover, there is also a unified treatment using log depth map for both orthographic projections and perspective projections Quéau et al. (2018a); Durou & Courteille (2007).

### 2.2 DEPTH EDGE DETECTION AND PRESERVATION

Depth discontinuities are the major barrier preventing a PDE solver from being a direct solution to real-world applications Durou et al. (2009). A weighting function is introduced as common approach to the modeling of depth discontinuities:

$$\min_z \int_\Omega w_u(\boldsymbol{n})E(\partial_u z - p(\boldsymbol{n})) + w_v(\boldsymbol{n})E(\partial_v z - q(\boldsymbol{n}))dudv, \tag{2}$$

where $w_u$ and $w_v$ are defined on a regular grid. Energy is naturally minimized when the weight $w_u$ and $w_v$ vanishing at the depth discontinuity naturally zeros out the constraint unnecessarily imposed upon the points located on the opposite sides of a depth discontinuity.

In addition to optimizers that directly suppress large numerical inconsistencies caused by violation of geometric constraint during their optimization processes Badri et al. (2014); Quéau & Durou

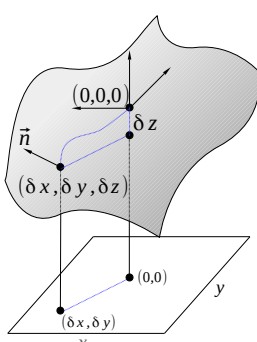 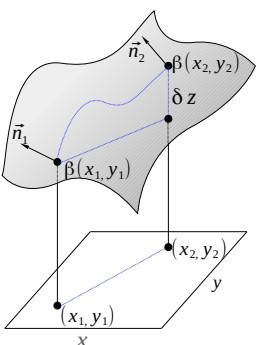 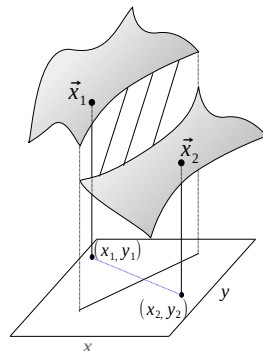

(a) a single point defines a differentiable neighborhood.

(b) two connected points located in a differentiable neighborhood.

(c) two points separated by a non-differentiable discontinuity.

Figure 2: A continuous field of polynomial coefficients $\beta$ represents a continuous surface in a point-wise manner. Each point $(x, y)$ is mapped to a local coordinate system whose origin $(0, 0, 0)$ coincides with the corresponding surface point $(x, y, z)$ where $z$ is to be determined (Figure 2a). Inside this local coordinates, the relative depth $\delta z$ of points in the neighborhood to the center is explicitly evaluated by $\beta$ according to Equation 3. $\beta$ can be estimated by relating two arbitrary points coexist in a differentiable neighborhood, as one point's polynomial can be used to evaluate a neighboring point's normal, and vice versa (Figure 2b). If one surface normal is directly observable, then $\beta$ can be learned. However, if the path connecting these two points on the surface is non-differentiable, resulting an invalid constraint that should be removed from training $\beta$. Thus, the location of depth discontinuities needs to be identified in the $xy$-plane (Figure 2c) in terms of edge weighting.

(2015), the process of devising a weighting function involving depth edge detection can be either static or dynamic. A static process detaches edge detection and surface depth estimation into two independent processes. Edge detection can be delivered by directly analyzing normal map Wu & Tang (2006), handcrafting Xie et al. (2019) or photometric cues Wang et al. (2012), etc.. As the first step towards integration over scattered normals, our work belongs to this category.

On the other hand, a dynamic process relies on depth estimation online and updates the weighting function iteratively, assuming weighting function to be in the form $w(\boldsymbol{n}, z)$. For example, one acn define an $\alpha$-surface, and at each iteration, gradients for connected neighbors (less than $\alpha$) are taken into account. Alternatively, one can also describe this process using anisotropic diffusion Quéau et al. (2018b). The most recent example involves applying semi-differentiable connectivity pattern to produce the weighting function, whereas the pattern itself is updated along with the online estimation of the surface depth Cao et al. (2022).

## 2.3 Polynomial Surface Fitting and Point Cloud Geometry

Per-point normal of a scattered point cloud can be estimated as a weighted average of neighboring surface normals Ben-Shabat & Gould (2020). The surface is parameterized using polynomials Cazals & Pouget (2005), based on which polynomial coefficients are fitted by least-square solver. The weight is per-point and learned as a product of supervised learning. This learning process can be improved by making the order and scale of the polynomial location-adaptive Zhu et al. (2021). Moreover, the implicit 0-level set can be integrated to further improve the global orientation consistency of the normal estimation for a compact surface Li et al. (2024). We extend these formulation to the inverse domain where the shape structure is not provided. This formulation can also be applied to shading analysis Xiong et al. (2014).

## 3 Continuous Surface Modeling

Our model follows the established n-jet model Cazals & Pouget (2005) to parameterize a locally differentiable surface, based on which we also present a globally consistent design that approximates a general surface in the presence of depth discontinuities.

## 3.1 N-Jet Surface Model and a Field of Polynomials

As illustrated in Figure 2a, a locally-differentiable surface is defined in a local coordinate system with its center coinciding the origin $(0, 0, 0)$. All surface points $(\delta x, \delta y, \delta z)$ satisfy a polynomial height function $J_n : \mathbb{R}^2 \to \mathbb{R}$ that maps the **local displacement** $(\delta x, \delta y)$ to **local relative height** $\delta z$ as follows:

$$\delta z = \sum_{k=0}^{m} \sum_{j=0}^{k} \beta_{k-j,j} \delta x^{k-j} \delta y^j, \tag{3}$$

where $m$ is the order of the polynomial and $\{\beta_j\}$ denotes a sequence of polynomial coefficients. Accordingly, surface normal $\boldsymbol{n}(\delta x, \delta y) = \frac{1}{\sqrt{z_x^2 + z_y^2 + 1}}(z_x, z_y, -1)$, where $z_x$ and $z_y$ are the first-order partial derivatives taken with respect to $x$ and $y$, respectively:

$$z_x(\delta x, \delta y; \boldsymbol{\beta}) = \sum_{k=0}^{m} \sum_{j=0}^{k} \beta_{k-j,j} \delta x^{k-j-1} \delta y^j \tag{4}$$

$$z_y(\delta x, \delta y; \boldsymbol{\beta}) = \sum_{k=0}^{m} \sum_{j=0}^{k} \beta_{k-j,j} \delta x^{k-j} \delta y^{j-1}$$

Since $\boldsymbol{n}(\delta x, \delta y)$ is parameterized by coefficients $\{\beta_j\}$, by providing sufficient number of observations for various $\boldsymbol{n}(\delta x, \delta y)$, $\{\beta_j\}$ can be estimated through linear regression estimation.

Since Equation 3 applies to a local neighborhood centering at an arbitrary point $(x, y)$, we are able to establish a **continuous** vector field of parameters, $\boldsymbol{\beta}(x, y)$, to describe the global shape in a consistent manner. This imposes a spatially symmetric constraint as illustrated in Figure 2b, where if two points, $(x_1, y_1)$ and $(x_2, y_2)$, are located on the same differentiable surface, their orientations can be reciprocally parameterized by each others' polynomial coefficients, imposing two spatial constraints between $\boldsymbol{\beta}(x_1, y_1)$ and $\boldsymbol{\beta}(x_2, y_2)$: (1) their gradient fields have to fit each others surface normal, $\boldsymbol{n}(x_1, y_1)$ and $\boldsymbol{n}(x_2, y_2)$, respectively; (2), the local relative height in Equation 3 between these two points is conserved.

Notation-wise, we apply Equation 3 to describe the entire field in terms of $(x, y)$ consistently as:

$$z(x, y)(\delta x, \delta y; \boldsymbol{\beta}) = z(\delta x, \delta y; \boldsymbol{\beta}(x, y)), \tag{5}$$

and we let $z_x(x, y)(\delta x, \delta y; \boldsymbol{\beta})$ and $z_y(x, y)(\delta x, \delta y; \boldsymbol{\beta})$ follow the same convention of notation.

## 3.2 Surface Model with Depth Discontinuities

Correctly identifying the location of depth discontinuities (Figure 2c) and properly utilizing this information for depth estimation is crucial for normal integration. The parameterization scheme proposed by bilateral normal integrator Cao et al. (2022) has demonstrated excellent performance on estimating depth map recovery for regular-grid measurements. We propose a decomposed surface parameterization to extend the benefit of this scheme to continuous domain. In particular, we categorize surface points into two groups: sample points $\boldsymbol{x}_s$ whose surface orientations are directly observable and grid points $x_g$ whose locations are aligned with their neighbors along either $x$-axis or $y$-axis.

Accordingly, this decomposed parameterization leads to two types of connections between two points: (1) grid-grid connection that links two adjacent grid points, which we denote as $\boldsymbol{x}_g$; (2) grid-sample connection that associates a grid point with one of its child sample points, denoted as $\boldsymbol{x}_s$, designated by the quadtree. As examplified in Figure 3a, bilateral normal integrator weighs two axis-aligned edges joining the same grid point by a binomial random variable whose outcomes are either $(0, 1)$ or $(0.5, 0.5)$, with $0$ indicating that the edge is completely cut off by a depth discontinuity and $1$ for complete connection, whereas $(0.5, 0.5)$ means the grid point is connected from the both sides with even balance. In addition, if we are also able to accurately estimate the normal map $N_g(x, y)$ consisting of all $\boldsymbol{x}_g$, the corresponding depth map $Z_g(x, y)$ can be obtained by a traditional normal integrator. Since a quadtree creates a rectangular "cell" that quarantines a sample point $\boldsymbol{x}_s$, it can be readily seen that the depth of a sample point $\boldsymbol{x}_s$ inside a cell relative to the cell's four vertices of $\boldsymbol{x}_g$, and the decomposed parameterization achieves a complete model for continuous surface.

# 4 CONTINUOUS SURFACE NORMAL INTEGRATION

As discussed in Section 3.1, our model interoperates with a network that establishes a mapping $\Phi : \mathbb{R}^2 \to \mathbb{R}^K$ that associates a coordinate $(x, y)$ with a sequence of $K$ polynomials coefficients $\{\beta_j\}$, leading to a continuous vector field $\boldsymbol{\beta}(x, y)$ encoding an point-wise n-jet surface whose local system of coordinates takes $(x, y)$ as its origin. Our model approximates this mapping $\Phi$ using a coordinate-driven neural network, which, through training, is expected to produce an accurate estimate normal map estimate, $N_g(x, y)$ on grid points.

In addition to learning $N_g(x, y)$, correctly identifying the locations of the depth discontinuities not only improves the accuracy of depth map for grid points $\boldsymbol{x}_g$ but also leads to a better formulation of the loss function which in turn improves the accuracy of $N_g(x, y)$. Since depth discontinuities are formulated on the premises that the structure of the surface is known, the training of the model essentially follows an iterative optimization process that alternatively refines $\boldsymbol{\beta}(x, y)$ and the likelihood estimate of the locations of depth discontinuities.

## 4.1 LEARNING GRID NORMAL MAP $N_g(x, y)$

Let $(\delta_x, \delta_y) = \boldsymbol{x}_s - \boldsymbol{x}_g$ indicate the displacement vector of $\boldsymbol{x}_s$ to $\boldsymbol{x}_g$. With a field of polynomial coefficients $\boldsymbol{\beta}(x, y)$ being parameterized by the network $\Phi$, we choose to train it by fitting both polynomials $\boldsymbol{\beta}_g(\delta_x, \delta_y)$ and $\boldsymbol{\beta}_s(0, 0)$ to $\boldsymbol{n}(\boldsymbol{x}_s)$ that is directly observable. Specifically, according to Equation 4, this leads to a design of loss function in terms of cosine distance function over grid-sample connections:

$$l_s = 1 - |\boldsymbol{n}(z_x(\boldsymbol{x}_s)(0, 0; \boldsymbol{\beta}_s), z_y(0, 0; \boldsymbol{\beta}_s)), \boldsymbol{n}(\boldsymbol{x}_s)|_{\cos},$$
$$l_{g,s} = 1 - |\boldsymbol{n}(z_x(\boldsymbol{x}_g)(\delta_x, \delta_y; \boldsymbol{\beta}_g), z_y(\delta_x, \delta_y; \boldsymbol{\beta}_g), \boldsymbol{n}(\boldsymbol{x}_s)|_{\cos}, \tag{6}$$

which is sufficient to train a surface that is differentiable over the entire field with sufficient observed samples $\boldsymbol{x}_s$.

Following a similar routine, there are two ways to evaluate normal map $N_g(x, y)$ using Equation 3. One is to evaluate the center of a differentiable neighborhood defined by $\boldsymbol{\beta}_g(0, 0)$, namely, obtaining the coefficients at $\boldsymbol{x}_g$ and evaluate the corresponding polynomials at $(0, 0)$. Alternatively, it is also possible to evaluate an adjacent but overlapping neighborhood centered at a nearby sample point $x_s$ as $\boldsymbol{\beta}_s(-\delta_x, -\delta_y)$, which traces the displacement from $\boldsymbol{x}_g$ back to $\boldsymbol{x}_s$. To sum up, according to Equation 4 and 5, $N_g(x, y)$ evaluated at point $\boldsymbol{x}_g$ or $\boldsymbol{x}_s$ in its vicinity can be read as:

$$\boldsymbol{n}(\boldsymbol{x}_g) = \frac{1}{\sqrt{z_x^2 + z_y^2 + 1}} \{z_x(\boldsymbol{x}_g)(0, 0; \boldsymbol{\beta}_g), z_y(\boldsymbol{x}_g)(0, 0; \boldsymbol{\beta}_g), -1\} \tag{7}$$

as well as:

$$\boldsymbol{n}(\boldsymbol{x}_g) = \frac{1}{\sqrt{z_x^2 + z_y^2 + 1}} \{z_x(\boldsymbol{x}_s)(-\delta_x, -\delta_y; \boldsymbol{\beta}_s), z_y(\boldsymbol{x}_s)(-\delta_x, -\delta_y; \boldsymbol{\beta}_s), -1\} \tag{8}$$

It is worth noting that, grid-sample connections are determined by the topology assigned by a quadtree, and not all loss functions derived are valid in the presence of depth discontinuities. Therefore, we assign a weight for each connection and the loss function should be read using Equation 6 as:

$$L = \sum_s l_s + \sum_{s,g} w_{s,g} l_{s,g}, \tag{9}$$

and we model $w_{s,g}$ using a set of binomial random variables derived from the topology from the existing design of bilateral normal integration using the learned $N_g(x, y)$.

## 4.2 ESTIMATING GRID DEPTH MAP $Z_g(x, y)$ FROM $N_g(x, y)$

Bilateral normal integration Cao et al. (2022) models two axis-aligned grid-grid connections joining the same grid point with a single binomial random variable. We port this parameterization to our estimation routine of the grid depth map $Z_g(x, y)$ based on a learned $N_g(x, y)$. In particular, each

grid point $\boldsymbol{x}_g$ shares four connections with its four neighbors, which are assigned with a weight denoted as $w_{x+}$, $w_{x-}$, $w_{y+}$, and $w_{y-}$, respectively. These four quantities measure the outcomes drawn from two independent binomial random variables as $w_{x+} + w_{x-} = 1$ and $w_{y+} + w_{y-} = 1$. Moreover, they are parameterized by comparing the depth differences across the connection pair: $w_{x+} = \sigma(\delta_{x-}z - \delta_{x+}z)$ where $\sigma(\cdot)$ denotes the Sigmoid function.

Moreover, since $N_g(x, y)$ contains grid points with non-uniform spacing, our solver directly utilizes four connection-wise orthogonality constraints for each grid point at the center as follows:

$$
\begin{aligned}
(n_x \delta_{+}x + n_z \delta_{x+}z)w_{x+} &= 0 \\
(n_x \delta_{-}x + n_z \delta_{x-}z)w_{x-} &= 0 \\
(n_y \delta_{+}y + n_z \delta_{y+}z)w_{y+} &= 0 \\
(n_y \delta_{-}y + n_z \delta_{y-}z)w_{y-} &= 0,
\end{aligned}
\tag{10}
$$

where $(n_x, n_y, n_z)$ denotes one measurement in $N_g(x, y)$ and $\delta_{+}x$, $\delta_{-}x$, $\delta_{+}y$ and $\delta_{-}y$ are explicitly fed to the solver. $\delta_{x+}z$, $\delta_{x-}z$, $\delta_{y+}z$ and $\delta_{x-}z$ are obtained by applying the corresponding directional difference operator to the unknowns $Z_g(x, y)$. Consequently, assembling these per-point four conditions together and transforming them to a minimum energy problem leads to a sparse symmetric linear system from which $Z_g(x, y)$ can be solved by a standard conjugate gradient method. Additionally, weight of grid-grid connections are alternatively updated alongside $Z_g(x, y)$.

We utilize the weight of grid-grid connections $w_{x+}$, $w_{x-}$, $w_{y+}$ and $w_{y-}$ obtained through bilateral normal integration to derive grid-sample connections $w_{s,g}$ for the loss function defined in Equation 9. To this end, as illustrated in Figure 3b, we divide the vicinity of each $\boldsymbol{x}_g$ into four quadrants, each of which is partitioned by the four grid-grid connections, respectively. We evaluate a weight of each quadrant to be the product of the associated grid-grid connection weight, and assign this weight to all sample points located in the same quadrant. Essentially, the product of two independent binomial random variables means that $x$-axis and $y$-axis depth-discontinuities take place independently, and its outcomes should also be 0-1, indicating the probability of a sample point $\boldsymbol{x}_s$ being connected with the center $\boldsymbol{x}_g$.

### 4.3 ESTIMATING CONTINUOUS SURFACE DEPTH FROM $Z_g(x, y)$

Continuous surface depth estimation also utilizes grid-sample connections over which relative heights of $\boldsymbol{x}_s$ to a set of nearby grid points $\boldsymbol{x}_g$ are integrated to produce a single depth value. This requires rearranging grid-centered grid-grid connections as four orthogonal boundaries for a sample-centered rectangular cell. Specifically, each sample point, $\boldsymbol{x}_s$, is circumscribed by these four boundaries, and we establish four grid-sample connections between the sample point and its four vertices made of grid points $\boldsymbol{x}_g$.

Because the grid-grid connections in the same cell are drawn from different binomial variables, normalization is required to produce a consistent estimate. Our solution is to formulate a binary clustering problem for each connection independently. In particular, a sample point $\boldsymbol{x}_s$ in a cell is to be evaluated against each of the four boundaries, where the connection over each boundary encodes one or two clusters with their centers located at its two ends. Here the weight of grid-grid connection serves as the prior distribution of the clusters. For instance, a grid-grid connection with 0 weight indicates that its two ends belong to two separated clusters; on the other hand, there exists a single cluster if the weight is greater than $0.5$. Quantitatively, we use cosine distance between the surface normal of the two ends of a grid-sample connection to measure likelihood of the sample point $\boldsymbol{x}_s$. Hence, normalized per-connection weight of grid-sample connection shall be obtained by maximizing the following likelihood function:

$$
l(w_{sg}) = |\boldsymbol{n}_{g1}, \boldsymbol{n}_s| w_g w_{sg1} + |\boldsymbol{n}_{g2}, \boldsymbol{n}_s| (1 - w_g) w_{sg2},
\tag{11}
$$

where $w_{sg1} + w_{sg2} = 1$, $|\cdot, \cdot|$ indicates cosine distance and $w_g$ is the boundary weight from one of $w_{x+}$, $w_{x-}$, $w_{y+}$ $w_{y-}$ through regrouping. Accordingly, a per-connection depth for $\boldsymbol{x}_s$ is evaluated as:

$$
z(\boldsymbol{x}_s) = z_{sg1} w_{sg1} + z_{sg2} w_{sg2}.
\tag{12}
$$

Furthermore, we evaluate Equation 12 for each connection in a cell and average the four readings out to produce the final estimation for $z(\boldsymbol{x}_s)$. This process is presented in Figure 3c.

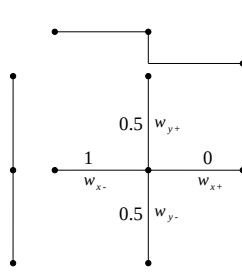 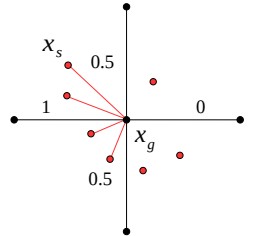 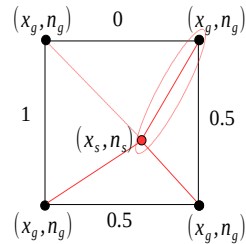

(a) grid-grid connection.

(b) grid-centered grid-sample connection.

(c) sample-centered grid-sample connection.

Figure 3: Axis-aligned grid-grid connection (Figure 3a) is delivered by bilateral normal integration Cao et al. (2022). Each grid point is 4-connected, with two neighbors along the $x$-axis and two along the $y$-axis. The connections are modeled in pairs in terms of edge weight denoting outcomes of a binomial random variable. When the path between these two points are differentiable, the corresponding weight is $(0.5, 0.5)$ indicating connectivity, and when a depth discontinuity cut in vertically on one axis, the weight is $(1, 0)$ with $0$ indicating dis-connectivity. To estimate normal $n_g$ (Figure 3b), we correlate the corresponding $x_g$ with its neighboring sample points $x_s$, where connection weighting is derived from the weight of grid-grid connection of $x_g$ (Section 4.1). To associate $x_s$ with its four neighboring cell vertices (Section 4.3) for depth prediction, we cluster these points according to cosine distances between their respective surface normal, where the number of clusters is derived from the re-grouped grid-grid connections (e.g. an edge weight of $0$ implies the existence of two separate clusters). The depth of sample $x_s$ is the average of its relative depth to each cell boundary offset by the grid depth, $z_g$, previously obtained.

Finally, because surface normal can be estimated for an arbitrary query point in the place of $n_s$, the proposed design can be extended to depth estimation for an arbitrary sample point with unobserved orientations, hence this routine generalizes to a continuous surface.

## 5 EXPERIMENT

We prepare two types of input data to our model to test its effectiveness. We first conduct experiment using input data stored on a standard regular grid to test verify its "backward compatibility". Namely, we do not assume the structure of data is known as a priori and the locations of grid points are determined by a quadtree independently. In this case, we compare our results against results obtained from existing counterpart normal integration approaches and let them process grid input directly. Throughout our experiment, we fix the order of per-point polynomials to be 3, meaning the length of coefficient sequence is constant 9 (e.g. Equation 3 contains 9 additive terms).

In addition, we also prepare a set of scattered measurements in the form of 5-tuple, $(x, y, n_x, n_y, n_z)$, as exemplified in Figure 1, to which existing designs do not apply. To achieve fair comparison, we follow a similar routine to our design that applies a quadtree to prepare a best-effort grid based setting that is suitable for counterpart method to deliver meaningful results. In particular, we establish the grid-based normal map through nearest neighbor value mapping from the unstructured input data. We normalize measurement coordinates to be inside a $[-1, 1] \times [-1, 1]$ square. Our solution is implemented using PyTorch Paszke et al. (2017), and experiments are conducted on a single Nvidia RTX 4090 GPU with 24G RAM. K-Nearest-Neighbor search is performed by Pytorch3D Ravi et al. (2020). Estimation results are evaluated using Mean Absolute Depth Error (MADE).

### 5.1 BENCHMARK DATA

Two data sets are used in our experiments: DiLiGenT Shi et al. (2016) containing 9 models and ground truth normal map with its multi-view version DiLiGenT-MV Li et al. (2020) and Sculpture Fouhey et al. (2016). In particular, DiLiGenT-MV provides the ground truth shape of 5 of 10 models originally contained in DiLiGenT, which we adopt for quantitative comparison, and we process the

|  | bear | cow | buddha | reading | pot2 |
|---|---|---|---|---|---|
| ours (sample) | 14.40 | 47.37 | 31.73 | 25.75 | 29.09 |
| ours (grid) | 20.09 | 31.24 | 18.79 | 27.26 | 31.35 |
| IPF Cao et al. (2021) | 24.29 | 38.66 | 8.23 | 33.59 | 17.35 |
| BiNI Cao et al. (2022) | 0.74 | 19.84 | 2.07 | 14.73 | 11.55 |

Table 1: Error of depth estimation evaluated in MADE as normal maps of DiLiGenT-MV is taken as input in grid representations. Our method (grid) performs inferior to BiNI but delivers comparable to IPF. Moreover, we also include the results obtained using sample queries drawn from continuous domain, it can be readily seen that our method is **insensitive** to input data representations and delivers reasonably good results in both cases. The geometric interpretation MADE is visualized in Figure 4.

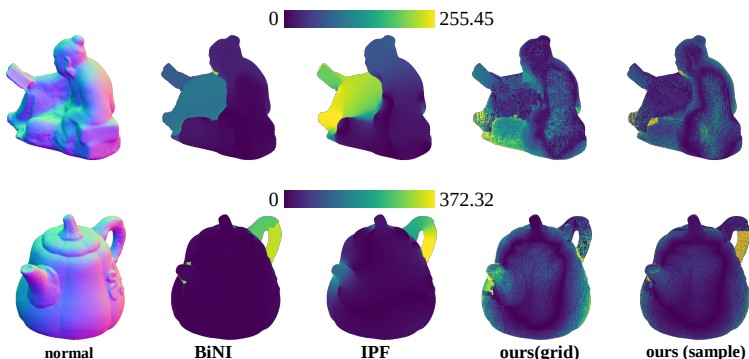

Figure 4: Estimation error of "reading" and "pot1". Despite the numerical variance of MADE obtained for different methods summarized in Table 1, shape geometry in general is captured and correctly estimated by our method. See appendix for results of more models.

other 4 for direct visual comparisons. The sculpture dataset provides direct ground truth shape in mesh, with which we use a renderer Yu et al. (2023) to generate the corresponding ground truth normal map from an arbitrarily selected angle. Moreover, we apply Halton sampler Berblinger & Schlier (1991) to draw random samples from the ground truth normal map to obtain the unstructured input in the form of 5-tuple.

## 5.2 BENCHMARK METHODS

Two recent approaches, Bilateral Normal Integration (BiNI) Cao et al. (2022) and inverse plane fitting (IPF) Cao et al. (2021), are used for comparison. In particular, BiNI delivers the state of the art performance, and it is worth mentioning that a variant of its parameterization of depth discontinuities is integrated into our workflow.

## 5.3 NORMAL INTEGRATION OVER REGULAR GRID

The numerical reconstruction error consuming grid-based input from DiLiGenT-MV dataset are tabulated in Table 1, and the results are visualized in Figure 4.

## 5.4 NORMAL INTEGRATION OVER CONTINUOUS DOMAIN

The last two columns of Figure 4 show that our solution is insensitive to the structure of input representation, be it of grid representation or generated from random samples. Moreover, Table 2 tabulates the estimation error of our methods and BiNI when processing the query input sampled from models in sculpture dataset, and the results are also visualzied and compared in Figure 5.

|  | head | bust | statue | skeleton |
|---|---|---|---|---|
| ours (sample) | 38.10 | 64.00 | 35.20 | 50.34 |
| BiNI Cao et al. (2022) | 120.04 | 183.59 | 70.94 | 101.01 |

Table 2: Error of depth estimation evaluated in MADE as unstructured depth queries of Sculpture are randomly-sampled. The results are visualized in Figure 4.

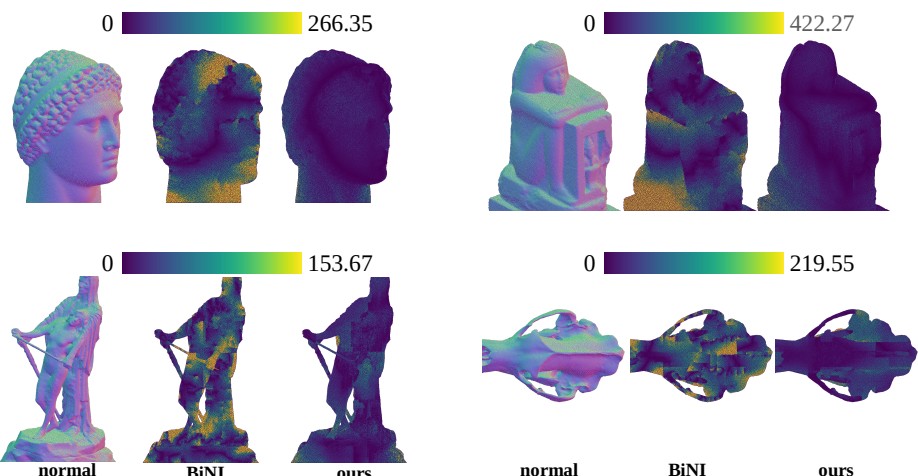

Figure 5: Estimation error of sculpture data set with randomly-sampled input. From left to right: randomly-sampled input normal in the form of 5-tuple query, estimation error of BiNI, estimation error of ours. It can be seen that our results outperforms BiNI when integrating samples drawn from a continuous surface.

It can be seen from these comparisons that when the representations of input are unstructured and randomly sampled, BiNI taking input with grid representation through nearest neighbor matching delivers an apparently inferior result to the results produced by our method. This shows that learning an accurate continuous gradient field is crucial for normal integration as there are spatially high frequency variations that cannot be captured by direct nearest neighbor matching, no matter how dense the grid points are distributed. Instead, performance gain can be achieved by accurately modeling the local geometry of the surface and precisely capturing the correlation between two arbitrarily located surface points.

## 6 CONCLUSION AND FUTURE WORK

This paper introduces a novel computational framework that, by taking coordinate-based depth queries, allows for normal integration to be performed over a continuous domain. We propose to represent continuous surface using a continuous field of learnable polynomial coefficients, and we integrate a depth-discontinuity-aware edge weighting scheme for pair-wise point connections into our training pipeline to obtain these parameters. Our experiment on various settings and various datasets shows that our method not only performs as effectively as existing approaches on traditional grid-based input, but also successfully delivers continuous surface normal integration that existing methods cannot handle. Furthermore, because continuous monocular surface representations enables flexible across-view alignment in a multi-view setting, extending this monocular design to multi-view explicit dense surface reconstructions is the goal of our future work.

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

## A  DIFFERENTIABLE LOCAL SURFACE AND LAPLACE SYSTEM

We set up a system of linear equations according to the first fundamental form of differential geometry. Specifically, a locally differentiable surface can be modeled by a linear Laplace system. In the presence of depth discontinuities, a weighting function is adopted evaluating pair-wise connections between two points, as indicated in Equation 10.

We assume each data point represents a discrete sample drawn from a continuous surface. As an example illustrated in Figure 6, points in a differentiable neighborhood are geometrically related by their pair-wise distances to the center point. Essentially, this configuration defines a $K$-connected graph with $N$ vertices and $KN$ directed edges. Correspondingly, we can setup a sparse $KN$-by-$N$ matrix, whose $i$-th row contains only 1 and $-1$ pair, with the corresponding column numbers indicating the head node and the tail node of directed edge $i$. Notably, when being applied to a regular grid, $D$ represents a numerical difference operator. In either case, this representation leads to:

$$Dz(x) = \delta_{\mathbf{z}}, \qquad (13)$$

where $z(x)$ is $N$-by-1 vectorized depth field, and $\delta_{\mathbf{z}}$ is a $KN$-by-1 vector whose entry indicates the depth difference between the vertices of each edge. With a local coordinate system whose origin $x_0$, and if the underlying surface is smooth, the first fundamental form in differential geometry dictates

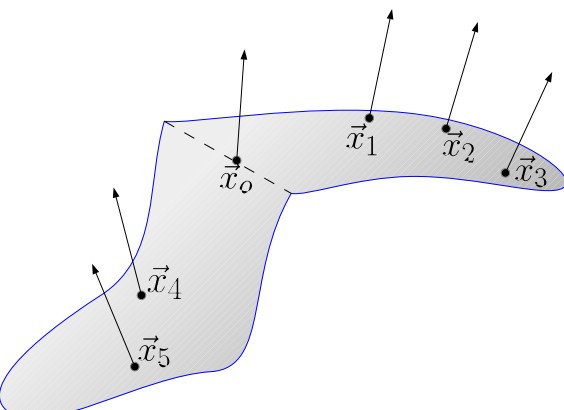

Figure 6: Sample points are arbitrarily drawn from a locally differentiable surface. Each point is associated with its neighbors, whose geometry is modeled by a polynomial according to Equation 3. This represents one of $K = 5$ conditions on pair-wise depth difference $\delta z_i$ binding neighbor $\vec{x}_i$ with center $\vec{x}_0$ imposed by the Laplace system of Equation 16. Because of the existence of depth discontinuity, $\vec{x}_4$ and $\vec{x}_5$ have weak connections with $\vec{x}_0$.

a linear approximation for it. In particular, any point $x_i$ in the neighborhood of $x_0$ can be expressed in terms of its normal $\boldsymbol{n}_0 = (n_x^0, n_y^0, n_z^0)$ as:

$$\boldsymbol{x}_i \approx \boldsymbol{x}_0 + (1, 0, \frac{n_x^0}{n_z^0})\delta_x^i + (0, 1, \frac{n_y^0}{n_z^0})\delta_y^i, \tag{14}$$

where $\delta_z^i$ can be expressed as:

$$\delta_z^i n_z^0 \approx -\delta_x^i n_x^0 - \delta_y^i n_y^0, \tag{15}$$

which can be rearranged and reduced to Equation 10.

In practice, numerical instability often arises when Equation 13 is solved directly for $D$ being too sparse. A remedy is to instead equate the distance of $\boldsymbol{x}_0$ to the plane spanned by its differentiable neighborhood containing $\{\boldsymbol{x}_0, \ldots, \boldsymbol{x}_k\}$. This amounts to performing a normalized contour integral around $\boldsymbol{x}_0$ and in discrete domain this is done by multiplying both sides by $D$ and normalized by the corresponding node degrees. In other words, Equation 13 extends to:

$$D^T N_z^{-1} D\boldsymbol{z}(\boldsymbol{x}) = D^T N_z^{-1}\boldsymbol{b} \tag{16}$$

where $\boldsymbol{b}$ is a $KN$-by-1 vector whose each entry evaluates the RHS of equation 15 for an edge and $N_z^{-1}$ is a $KN$-by-$KN$ diagonal matrix whose non-zero entry contains the value of the corresponding $n_z^0$. It is worth noting that, by equating $\delta_x$ and $\delta_y$ over the entire domain, we can apply the same interpretation to derive its regular grid counterpart in the form of a minimal energy formulation. In short, $D^T D$ represents the Laplacian matrix of a graph discretizing a smooth manifold and $D$ defines the local geometric structure of the surface.

## B NETWORK ARCHITECTURE

Figure 7 presents the architecture of the network proposed by our design. The network maps the coordinates $(x, y)$ to a sequence of coefficients $\beta$ of 9 entries. This means to evaluate the points $(x + \delta x, y + \delta y)$ in the neighborhood of $(x, y)$ as $z(\delta x, \delta y)$ being evaluated according to Equation 3 using the obtained $\beta$.

## C LOSS FUNCTION

We fit directly the gradient of to surface $z(x, y)$ to the corresponding normal as:

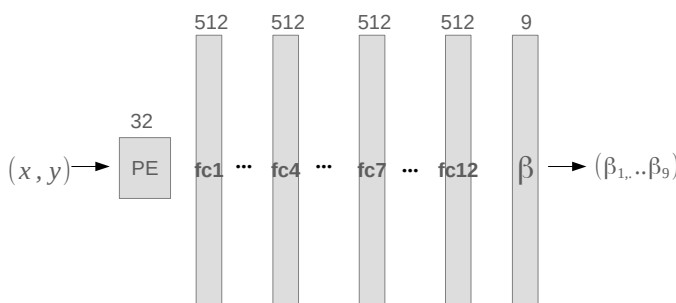

Figure 7: The architecture of the network that approximates the mapping from $(x, y)$ coordinates to a sequence of coefficients $\beta$ of 9 entries.

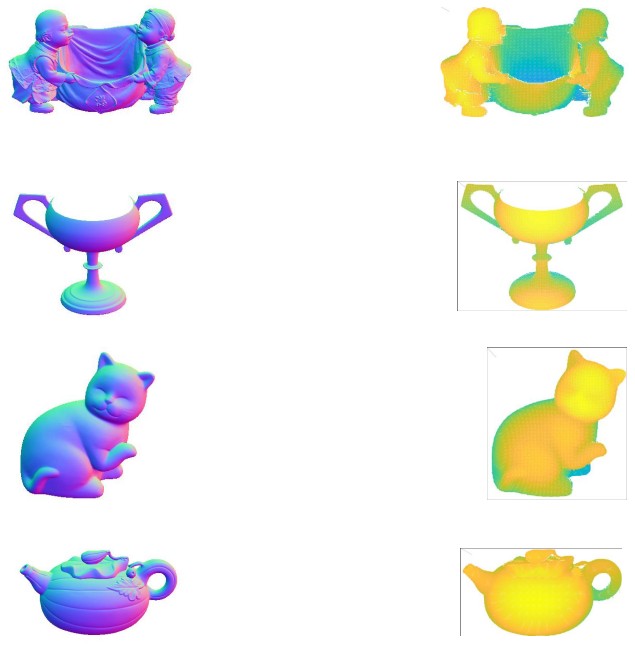

Figure 8: DiLiGenT. From left to right: input normal map of grid representation, estimated point cloud produced by our method.

$$L(\boldsymbol{z}_x, \boldsymbol{z}_y, \boldsymbol{n}) = (z_x n_z + n_x)^2 + (z_y n_z + n_y)^2 + (n_z \sqrt{z_x^2 + z_y^2 + 1} - 1)^2 \tag{17}$$

where $\boldsymbol{n} = (n_x, n_y, n_z)$ is observable and $z_x$ and $z_y$ are in terms of learnable $\beta$.

## D    RESULT

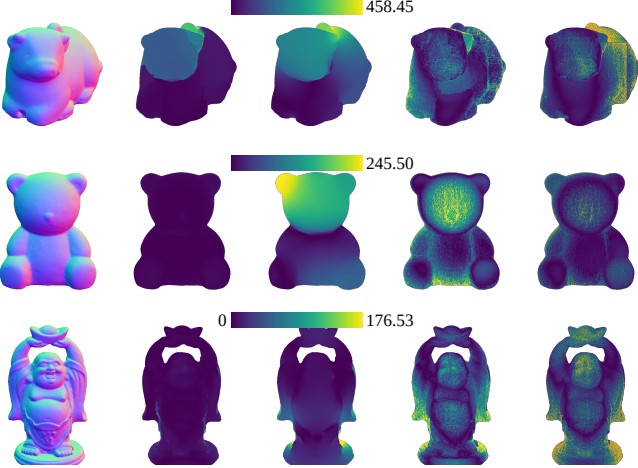

Figure 9: DiLiGenT. From left to right: input normal map of grid representation, error map produced by BiNI, IPF, our method and our method taking unstructured input sampled from the same model.

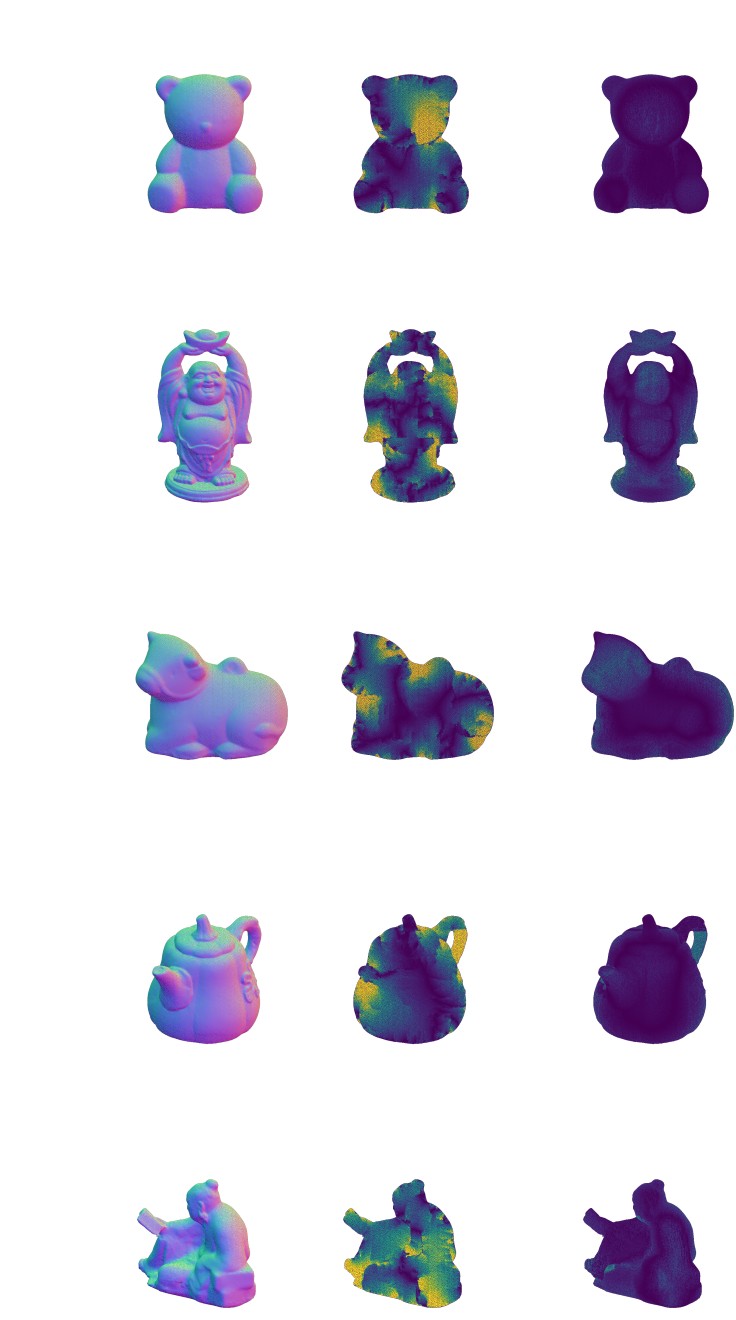

Figure 10: DiLiGenT. From left to right: randomly-sampled input normal , per point error produced by BiNI and our method.

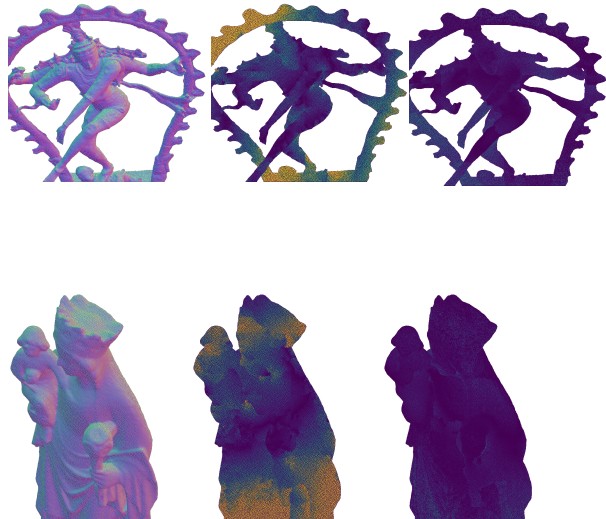

Figure 11: Sculpture. From left to right: randomly-sampled input normal , per point error produced by BiNI and our method.

