# OpenReview forum: "Continuous Surface Normal Integration"
_ICLR.cc/2025/Conference — ICLR 2025 Conference Withdrawn Submission_

### Official Review · Reviewer_HpWf · 2024-10-20

**Soundness:** 2
**Presentation:** 1
**Contribution:** 2
**Rating:** 3
**Confidence:** 4

**Summary:**

this paper addresses the normal integration problem under the assumption of (a) a set of points with unknown orientation and (b) a set of points where the normal is given. (a) are in a grid structure and (b) are random samples around (a).
The author proposes to model the surface using a higher-order polynomial and the normal can be naturally represented by a polynomial as well, as the derivative of the surface.
The author proposes to simultaneously estimate the depth of points as well and model the depth discontinuity using the constraint that points in a continuous surface, their normals can be represented by other each's polynomial coefficients.

**Strengths:**

1. higher-order polynomial surface representation is novel (order >= 2).
2. simultaneously estimate surface normal, depth, and depth discontinuity make sense.
3. The authors propose a network to estimate continuous surface normals, which is useful.

**Weaknesses:**

One of the biggest problems of the paper in my opinion is the writing. Unfortunately, I found the paper really hard to follow. I tried my best to understand the method and the experiment procedure that the authors wanted to present. But I still have a lot of unclear points.

I listed some of the questions and I'll update this part after hearing from the author.
I put the rating for reject for now since too many unclear part but I'm happy to hear the rebuttal from the author.

**Questions:**

1. The authors assume input (or the training data) is a set of points with known normals and non-regular-grid points where the normal is to be estimated. Under what circumstances one could have a situation like this? I can imagine one has a set of depth images that gives you unstructured points with normal (estimated from depth images). how do you obtain the non-regular grid points?
2. the authors claim to estimate continuous normal on the surface and normals are only defined on the surface, does it mean that the grid points are located on the surface and the methods can only estimate the normal of the grid points? Then what about the points off the grid?
3. equation (4) wouldn't it be something like $z_x = \sum \sum \beta_{k-j,j}(k-j) \delta x^{k-j-1} \delta y^j$ or did I misunderstand something?
4. I get the part that if two points are on a continuous surface, they should be represented by each other's polynomial coefficients, but maybe the author can elaborate more on how and where you used it as a depth discontinuity indicator. Especially how do you check if they cannot represent each other?
5. How do you get the edge weight? I'm a bit confused if it is gotten from Line096 you mentioned the connectivity or from the depth discontinuity estimation. Which latter one makes more sense but how?
6. I in general don't understand the experiment setting. Maybe the authors can explicitly explain what is the input and output of your experiments and what effect you would like to show.

---

### Official Review · Reviewer_3Tkr · 2024-10-28

**Soundness:** 2
**Presentation:** 3
**Contribution:** 2
**Rating:** 5
**Confidence:** 3

**Summary:**

The paper presents a method for integrating normal observations into continuous depth maps. The method first defines a quadtree structure over the input image and then tries to fit the network to the beta parameter for each pixel. The beta parameter is a parametrization of the n-jet representation which defines the continuous surface. By solving for the best observation whose derivative reflects the normal map, the method can recover accurate the continuous depth map from the given normal structure point cloud. The method is compared against IPF and BiNI, and shows robustness when the point cloud is provided either with a grid format or unstructured format. Qualitative evaluations also show the superiority of the method.

**Strengths:**

1. The paper is well-written and the technical details are presented clearly. Essential background knowledge is introduced (including the PDE solver and the polynomial surface fitting) and this provides the readers enough information to understand the rest of the pipeline.

2. The paper is mathematically sound, with all the notations properly defined and all the equations looking reasonable to me.

3. The method seems to be able to generate decent results in comparison to the baselines in terms of both quality and numbers.

**Weaknesses:**

1. The quality of the results does not look very impressive to me. Given the high quality of the input normal map and the complexity of the pipeline, I would expect a better-quality depth map with more details and sharp transitions. From both the tables (Table 1) and the figures (Figure 4) I could not see a clear distinction or superiority between the proposed method vs existing baselines. It seems that the proposed method not only has higher errors in some cases but also exposes a more severe problem of outliers, as evidenced by the very large error values in the depth map (e.g. 255.45, 372.32, etc.).

2. The evaluation of the method could be improved. Right now the method is only compared against baselines on a contrived dataset with few data samples, and this does not reflect the true performance when applied to larger datasets or scenarios. It would be advisable that the authors curate a dataset containing at least thousands of examples and compute the average error. Such a dataset would better contain simulated noise to reflect the property of real-world captures. In the meantime, I feel that the task of recovering depth from depth maps might be better tackled (or initialized) with a state-of-the-art depth estimator. For example, I tried to directly feed the image of the normal maps into the DepthAnythingV2 network (https://huggingface.co/spaces/depth-anything/Depth-Anything-V2), it could already give me very decent results. Such analysis and comparisons with modern deep-network-based estimators are necessary.

3. The computation required to compute the depth map from the normal map is missing. It would be nice if the authors could provide details on the computation time, hardware, and the memory taken to reconstruct the samples. Depending on the number of input points or the image resolution, does the processing time/required compute vary a lot?

4. The method introduces many components, but there is no ablation study on which part is important, and which hyperparameter affects the results the most. It is hence hard to nail down the core contribution of the paper.

**Questions:**

I don't have major questions after reading the paper at this moment.

---

### Official Review · Reviewer_k71w · 2024-11-02

**Soundness:** 3
**Presentation:** 3
**Contribution:** 1
**Rating:** 3
**Confidence:** 3

**Summary:**

This paper presents a method for surface normal integration. Unlike the existing methods considering (regular) grids as the normal coordinates, the proposed method deals with the continuous domain for the surface normals. To achieve this, the proposed method uses coordinate-based MLP. The method also incorporates with a bilateral normal integration method (Cao et al. 2022). The experiments show that the method achieves comparable (or slightly worse) accuracies for regular-grid normal integrations. The proposed method may be useful for irregular normal samples.

**Strengths:**

### Using coordinate MLP for continuous surface normal integration
Using coordinate-based MLP for surface normal integration should be a reasonable idea. It naturally results in the continuous surface normal integration.

### First (single-image) continuous surface normal integration
The method is the first attempt using coordinate-based MLPs for normal integration of single-image surfaces.

**Weaknesses:**

### Using coordinate MLP for surface normal integration
On the other hand, the use of coordinate-based MLPs for surface normal integration is straightforward, and in multi-view normal integration setup, using those is recently common, for example, in SuperNormal [a]. In such a sense, [a] already achieves (although in a multi-view setup) continuous-domain surface normal integration.

[a] Cao et al., SuperNormal: Neural Surface Reconstruction via Multi-View Normal Integration, CVPR 2024.

### Experimental results
Experimental results show that by inputting normals with a regular grid (which seems to be a natural setup), the proposed method's performance is slightly behind existing methods.

**Questions:**

Readers should wonder what practical scenarios require the irregular normal input (i.e., normals not defined at regular grids).

---

### Official Review · Reviewer_MgUc · 2024-11-03

**Soundness:** 3
**Presentation:** 2
**Contribution:** 2
**Rating:** 5
**Confidence:** 3

**Summary:**

This paper addresses an important problem in computer vision: reconstructing depths from a single-view image based on normal mapping. To tackle this ill-posed inverse problem, the authors employ a neural network to parameterize the surface into polynomial height functions.

**Strengths:**

- The paper uses polynomial height functions to parameterize surfaces, which is a reasonable approach that renders the process differentiable and suitable for neural network learning.
- It enables the estimation of normals and depth calculations at any point on the surface, unlike BiNI (Cao 2021), which can only compute depth at points corresponding to image pixels and requires solving non-convex optimization problems.

**Weaknesses:**

- The paper lacks extensive discussion and comparison with BiNI, particularly in how BiNI handles discontinuity preservation. No results addressing this capability are presented in this study.

**Questions:**

The method proposed involves three stages: initially obtaining a continuous normal map, then using BiNI to derive depth at discrete points, and finally achieving continuous depth using a quadtree-based method along with normal information. My question is, why not directly use the depth from the polynomial surfaces? Comparing the depth and normals from the polynomial surfaces with the depth obtained from BiNI could potentially streamline and enhance network learning. Wouldn’t this simplify the learning process? Please correct me if I am mistaken.

The paper presents experimental results using heat maps to reveal depth differences from the ground truth. To better demonstrate the model's accuracy, I suggest visualizing the reconstructed 3D models from different viewing angles.

---

### Note · Authors · 2024-11-14

I have read and agree with the venue's withdrawal policy on behalf of myself and my co-authors.